# Separating natural from human enhanced methane emissions in headwater streams

Yizhu Zhu [1], J. Iwan Jones[1], Adrian L. Collins[2], Yusheng Zhang[2], Louise Olde[1,2], Lorenzo Rovelli[3], John F. Murphy [1], Catherine M. Heppell[4] & Mark Trimmer [1✉]

Headwater streams are natural sources of methane but are suffering severe anthropogenic disturbance, particularly land use change and climate warming. The widespread intensification of agriculture since the 1940s has increased the export of fine sediments from land to streams, but systematic assessment of their effects on stream methane is lacking. Here we show that excess fine sediment delivery is widespread in UK streams ($n = 236$) and, set against a pre-1940s baseline, has markedly increased streambed organic matter (23 to 100 g m$^{-2}$), amplified streambed methane production and ultimately tripled methane emissions (0.2 to 0.7 mmol CH$_4$ m$^{-2}$ d$^{-1}$, $n = 29$). While streambed methane production responds strongly to organic matter, we estimate the effect of the approximate 0.7 °C of warming since the 1940s to be comparatively modest. By separating natural from human enhanced methane emissions we highlight how catchment management targeting the delivery of excess fine sediment could mitigate stream methane emissions by some 70%.

---

[1] School of Biological and Behavioural Sciences, Queen Mary University of London, London E1 4NS, UK. [2] Sustainable Agriculture Sciences, Rothamsted Research, North Wyke, Okehampton, Devon EX20 2SB, UK. [3] Institute for Environmental Sciences, University of Koblenz-Landau, Landau, Germany. [4] School of Geography, Queen Mary University of London, London E1 4NS, UK. ✉email: m.trimmer@qmul.ac.uk

A tmospheric methane reached its highest-ever concentration of 1900 ppb in 2021[1], calling for action to tackle all methane sources. Despite covering only a small fraction of the Earth's land surface, rivers and streams make a significant contribution to the global methane budget[2]. Rivers and streams actively transform organic matter into methane in their bed sediments[3,4] and, along with methane carried in from the catchment[5,6], emit some 31 Tg methane to the atmosphere every year, which almost nullifies the estimated soil methane sink[2,7]. Understanding the drivers of methane production in rivers and streams is of acute importance.

Across the globe, the water resources and biodiversity of the majority of the stream and riverine habitats are threatened by a multitude of anthropogenic disturbances[8]. The intensification of agriculture is a globally widespread threat[8] that degrades the quality of water courses by delivering excess fine sediment (<2 mm)[9–11]. Some sediment delivery is a natural component of flowing waters—shaping channel morphology and bed texture—and rivers and streams are natural sources of methane[12,13]. However, fine sediments generated by intensive agriculture in excess of natural baselines can, if deposited on the bed[14,15], reduce bed permeability[16] and restrict the flow of oxygen[17,18] to create an enriched habitat for methane production[19–21]. While it is widely recognized that stream and river methane concentrations can be elevated in agricultural catchments[12,22–24], there has been no systematic attempt to separate natural from human-enhanced methane emissions in rivers and streams. Projections suggest that our capacity to mitigate the off-site consequences of agriculture will be even more challenged under future climates. This is due to even higher elevated loss of sediment and nutrients driven by extreme rainfall events that will require significant land cover change (e.g., swapping crops for trees) to mitigate such externalities[25,26].

Methane emissions related to excess fine sediment may be particularly pronounced in headwater streams, as tending to be shallow with slower flowing water, they can facilitate the deposition of fine sediments[4,27]. Moreover, headwater streams, although often overlooked, make up 88% of the global stream length[28] and drain a substantial proportion of the land surface. Given that intensification of agriculture is widespread globally[29], headwater streams are commonly prone to ingress of excess fine sediment.

In addition to the landscape effects driven by anthropogenic disturbance, there is a need to partition the effects of warming from those driven by excess fine sediment in streams. For example, methanogenesis is sensitive to warming and its well-characterized temperature sensitivity of 0.93 eV[30] could potentially increase methane emissions by 1.7-fold under the strongest 4 °C warming scenario for the end of the twenty-first century[31,32]. Nevertheless, since the 1940s, the rapid intensification of modern agriculture has increased the export of fine sediments from catchments with a strong potential to increase organic matter on the beds of streams and rivers[11,19]. As the degradation of organic matter can provide substrates (acetate, carbon dioxide, and hydrogen) for methanogenesis[33], a process known to be substrate limited[34], the delivery of excess fine sediment has the potential to trigger a significant amount of human-enhanced production and subsequent emission of methane[20]. For example, short-term experimental additions of organic matter to lake and reservoir sediments enhanced their methane production potentials by threefold to 30-fold[35,36], which is considerably greater than the projected increase through warming. While correlations between methane (concentration or production) and either organic matter or temperature are recognized in freshwaters, how they systematically govern methane is poorly characterized, especially for streams[20,24,37–39]. Therefore, it is important to characterize the interaction between organic matter and temperature in order to separate the relative effects of organic matter associated with excess fine sediment and recent warming on methane emissions.

Here, we select 236 UK streams, with the majority (174) being small headwaters. We then reconstruct their delivery rates of fine sediment for the pre-1940s and use this as a modern natural baseline (see "Methods") to establish recent human-enhanced changes in methane emissions due to the intensification of agriculture since the 1940s. We demonstrate that excess fine-sediment delivery is not only widespread but that the associated increases in streambed organic matter have increased methane emissions too. In contrast, the estimated increase in methane production based on warming since the 1940s is comparatively modest. Our separation of natural from human-enhanced methane emissions suggests that real reductions in methane emissions are possible if well-targeted fine-sediment management strategies are applied in future.

## Results

**Widespread excess fine-sediment pressure and streambed organic matter.** Since the most dramatic modern increase in sediment yields occurred after the 1940s, when the intensification of agriculture took-off[11], we used pre-1940s sediment yields to establish a modern natural baseline (see "Methods"). This baseline reflects modern background rates of fine-sediment delivery that are suitable for sustaining a naturally healthy aquatic habitat in modern times[11]. In England and Wales, 236 streams were selected from agricultural catchments and the majority (74%) of these were small headwaters (174 streams with a catchment area <15 km², see Supplementary Table 1 for characteristics of the study streams) that are potentially more susceptible to any disturbance in their surrounding catchments. Furthermore, 80% of the study streams were receiving fine sediments in excess of the pre-1940s natural baseline and we categorized their magnitude of excess fine-sediment delivery relative to the baseline. Accordingly, 31 streams were under mild sediment pressure, receiving an average of 48-fold more fine-sediment delivery while the other 158 streams were under severe sediment pressure, receiving an average of 758-fold more fine sediment relative to the baseline (Fig. 1a, t-statistic, both $P < 0.001$).

As the amount of fine sediment deposited on a streambed reflects the balance between sediment delivery and the stream's capacity to transport and flush-out sediment[27], we standardized our estimates of excess fine-sediment delivery to specific stream power (Eq. (2)) and tested for any correlation with streambed organic matter (Fig. 1b, measured as ash-free-dry weight per m² (g m⁻²)). In line with our hypothesis, streambed organic matter was positively correlated with standardized excess fine-sediment delivery in those streams impacted by excess fine-sediment delivery (i.e., mild or severe excess fine sediment pressure, F-statistic, both $P < 0.001$), suggesting an increase in streambed organic matter due to excess fine-sediment delivery. Furthermore, this standardized measure of mild to severe pressure represents a 15-fold to 150-fold increase in fine-sediment delivery relative to the pre-1940s natural baseline (Fig. 1b, t-statistic, both $P < 0.001$).

Our standardized measure of fine-sediment delivery in natural baseline streams i.e., those streams with no excess relative to the pre-1940s, represents a natural hydrological baseline that we can use to separate out any agriculturally-induced excess of streambed organic matter. Accordingly, we used the relationship in Fig. 1b and Eq. 8 to back-calculate natural background streambed organic matter in the pre-1940s natural baseline. Prior to the intensification of agriculture, all our streams were receiving background rates of sediment delivery that produced natural variation in streambed organic matter from 6 to 77 g per m²

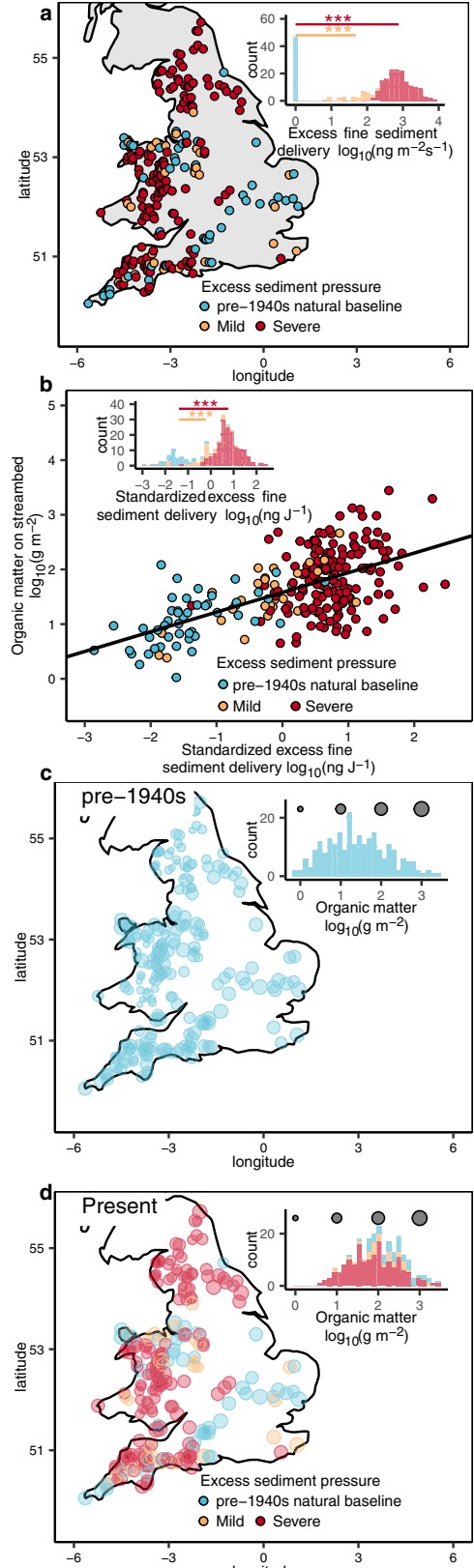

**Fig. 1 Excess fine-sediment delivery into streams increases organic matter on streambeds ($n = 236$ streams). a** The majority of streams that we sampled in England and Wales (80%, 189 out of 236 streams) are under mild to severe excess fine-sediment pressure, equivalent to 48-fold and 758-fold increases, respectively, in fine sediment relative to the pre-1940s natural baseline ($t$-statistic, ***: $P < 0.001$). **b** Standardizing excess fine sediment delivery to specific stream power, reveals a positive correlation between streambed organic matter and standardized excess fine-sediment delivery. Using the baseline streams as a natural hydrological baseline, the mild and severe pressure categories represent 15-fold and 150-fold increases in excess fine-sediment delivery ($t$-statistic, ***: $P < 0.001$). **c** Map of streambed organic matter reconstructed for the pre-1940s. **d** Map of streambed organic matter for the present day. Note, to illustrate the overall relationship between streambed organic matter ($y$ axis) and standardized excess fine-sediment delivery in each category[73], the $y$ axis data in **b** are presented as partial residuals while holding excess sediment pressure constant ($x$ axis, see Supplementary Table 3 for model selection and more details therein). The histograms give the distribution of data and the sizes of circles in panels **c** and **d** represent the estimated amount of organic matter per m$^2$ of streambed.

fourfold from 23 to 100 g per m$^2$ (Fig. 1d). In particular, streams under severe pressure have seen a sixfold increase in streambed organic matter (Eq. (9)), while those under mild pressure a 2.6-fold increase. Whereas it is perfectly natural for streambed sediments to produce methane, the strong substrate limitation of methane production in freshwaters[34] gives any excess organic matter the potential to increase streambed methane production[20,37] above the pre-1940s natural baseline.

**Effect of organic matter on the capacity and temperature sensitivity of streambed methane production**. Here, we were not only interested in how streambed organic matter, associated with excess fine sediment, increases methane production but also how organic matter interacts with temperature—as both have increased since the 1940s. To test this, we collected streambed sediments from another 14 streams in southern England and incubated them in temperature-controlled, laboratory microcosms (see Supplementary Fig. 1 for the stream locations). By plotting the natural-log-transformed rates of methane production of these incubations against standardized temperature, we were able to express the slope of the relationship, which represents the temperature sensitivity of production, as apparent activation energy (eV, Fig. 2a). Methane production was consistently sensitive to temperature across all streams (likelihood ratio test, $P = 0.57$), with an overall temperature sensitivity ($\overline{E_{MP}}$) of 1.1 eV (95% CI: 0.89–1.31), which is equivalent to a 1.8-fold increase in production per 4 °C (see Supplementary Table 4 for model comparisons).

Further, the intercepts (filled circles) in Fig. 2a represent a sediment's capacity to produce methane at a standardized temperature of 15 °C that we then used to isolate the effect of organic matter from temperature. In contrast to the highly conserved temperature sensitivity (i.e., the slopes in Fig. 2a), we observed substantial variation—~10,000-fold—in the capacity of our streambed sediments to produce methane (likelihood ratio test, $P < 0.001$, see Supplementary Table 4), from 0.001 to 68 nmol CH$_4$ g$^{-1}$ h$^{-1}$ (Fig. 2b).

Our observed wide variation in the capacity of sediment to produce methane was clearly related to its organic matter content (Fig. 2b). To formally test this, we fitted a linear model to the data (i.e., between the intercepts in Fig. 2a and each respective measure of organic matter) and found higher methane production with higher sediment organic matter (Fig. 2c, $P < 0.001$). Then, using the best fit linear model in Fig. 2c, along with the increase in streambed organic matter since the 1940s (23 g m$^{-2}$ to 100 g m$^{-2}$, median to median,

(first- and third quartile, respectively), with a median of 23 g per m$^2$ (Fig. 1c). Today with the vast majority of streams now under mild or severe pressure from excess fine sediment and a substantially reduced frequency of natural baseline streams, the range in streambed organic matter is 38 to 245 g per m$^2$ (first- and third quartile, respectively) and the median has shifted

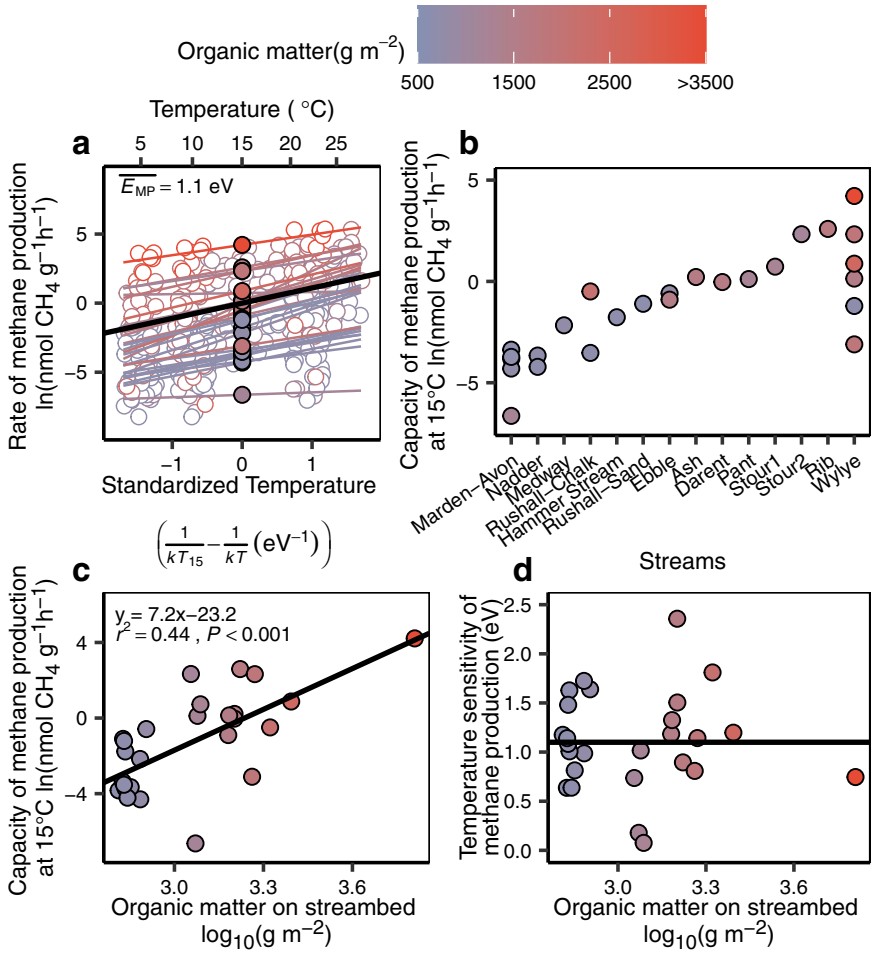

**Fig. 2 Effect of organic matter on both the capacity and temperature sensitivity of streambed sediments to produce methane ($n = 371$ observations in 14 streams). a** Plotting natural-log-transformed methane production against standardized temperature demonstrated a similar temperature sensitivity across all our streams and an average, overall sensitivity ($\overline{E_{MP}}$), of 1.1 eV (black solid line, 95% CI: 0.89–1.31). Filled circles represent the intercepts, i.e., the capacity of a sediment to produce methane standardized to 15 °C in each stream and the color represents the organic matter content of the sediment (g m$^{-2}$). **b** In contrast to the consistency in temperature sensitivity, the capacity of streambed sediments to produce methane (at 15 °C) varied by four orders of magnitude from 0.001 to 68 nmol CH$_4$ g$^{-1}$ h$^{-1}$ across our streams. **c** Methane production capacity (intercepts from panel **a**) is positively correlated with organic matter on streambeds ($t$-statistic, $P < 0.001$). **d** There was no relationship between the temperature sensitivity (slopes from panel **a**) of methane production and organic matter on streambeds ($t$-statistic, $P = 0.99$). The horizontal solid line represents the average temperature sensitivity of methane production in streams, i.e., 1.1 eV (95% CI: 0.45–1.75), in agreement with panel **a**. Therefore, while the temperature sensitivity of methane production was conserved, the potential for organic matter to exert variable and strong control over methane production was clear.

Fig. 1c, d), we can show that the capacity of our streambed sediments to produce methane has increased 100-fold through excess fine-sediment delivery due to the intensification of agriculture (Eq. (12)).

In contrast with the increase in methane production with sediment organic matter, no relationship was found between organic matter and the temperature sensitivity of methane production, i.e., between the slopes in Fig. 2a and each respective measure of organic matter ($P = 0.99$, Fig. 2d). In this additional analysis, the average temperature sensitivity of methane production across our streambed sediments, at 1.1 eV, was the same as the estimate provided by the analysis in Fig. 2a (mixed-effect model, see Supplementary Table 4), again corroborating that the temperature sensitivity of streambed methane production is highly conserved, whilst the variable effect of organic matter is pronounced and strong.

**Methane production with additional substrates.** Any organic matter on a streambed must first be fermented into simpler substrates before it can be used to produce methane[33], a process

that could occlude any interaction with temperature. Therefore, we experimentally tested the dependency between streambed methane production and substrates, and how they interact with temperature, by adding both immediate precursors to methanogenesis (acetate and hydrogen) and more complex, precursor substrates (betaine, trimethylamine (TMA), and propionate)[33]. Here we used sediments collected from a subset of 8 streams and repeated the laboratory incubations as before in Fig. 2a. In agreement with our initial analysis, the temperature sensitivity of methane production was the same in both control and substrate-amended sediments (likelihood ratio test, $P = 0.18$, see Supplementary Table 5 for model selection) and, at 1.0 eV (Fig. 3a, 95% CI: 0.90–1.16), was the same as for methane production fueled by natural variation in sediment organic content (1.1 eV, 95% CI: 0.89–1.31 vs. 1.0 eV, 95% CI: 0.90–1.16, Figs. 2a, 3a, respectively).

In contrast, all the substrates added significantly enhanced the capacity of the sediments to produce methane (the intercepts, represented as filled circles in Fig. 3a, b, significance tested by *post hoc* analysis, all $P < 0.001$): acetate, the direct substrate for acetoclastic methanogenesis[33], had the strongest effect, increasing

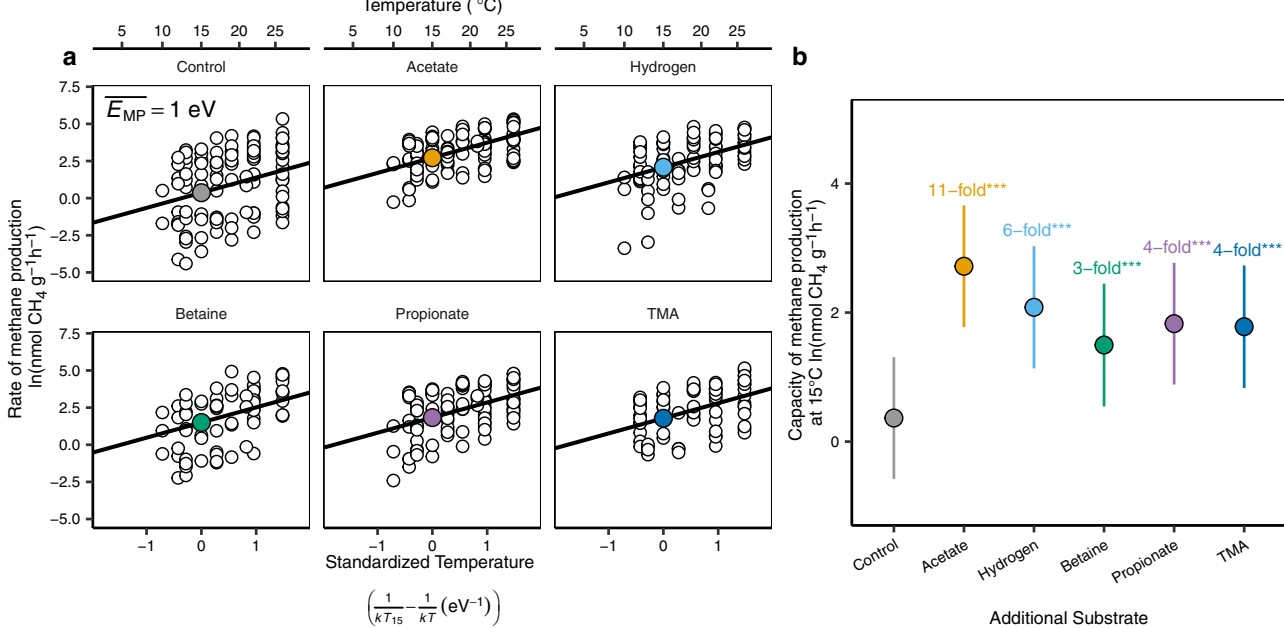

**Fig. 3 Temperature sensitivity and capacity of streambed sediments to produce methane with additional substrates ($n = 571$ observations in eight streams). a** The temperature sensitivity of methane production ($\overline{E_{MP}}$) was 1.0 eV (95% CI: 0.90–1.16) and was conserved both with and without additional substrates. **b** In contrast, all additional substrates enhanced the capacity of sediments to produce methane relative to unamended controls (*post hoc* analysis for pairwise comparison using Tukey method, ***: $P < 0.001$)—just as for the variation in streambed organic matter content in the field (Fig. 2b). The black solid lines in **a** represent the slopes, i.e., the temperature sensitivity, and the filled colored circles represent the intercepts i.e., the capacity of methane production standardized to 15 °C that are then reproduced for comparison in **b**. The vertical lines are 95% CI.

methane production by 11-fold compared with unamended controls, and higher than hydrogen (sixfold), propionate (four-fold), TMA (fourfold), and betaine (threefold). Clearly then, whether we observe the effect of natural variation in organic matter, or experimentally manipulate the availability of substrates for methanogenesis directly, the temperature sensitivity of methanogenesis is highly conserved whereas the effect of substrates is variable and strong.

**Excess fine-sediment delivery to streams and enhanced methane emissions**. As we had clearly identified the strong control that organic matter exerts over sediment methane production and that excess fine-sediment pressure has increased organic matter on UK streambeds, we expected that excess fine-sediment delivery would ultimately increase methane emissions from streams. To test this, we estimated methane emissions for a further subset of 29 streams selected from across our three categories of excess fine-sediment pressure (pre-1940s natural baseline, mild, severe) (see Supplementary Fig. 1) and standardized their emissions to specific stream power (Eq. (6)), just as we had done for excess fine-sediment delivery. In line with our expectation, methane emissions increased with excess fine-sediment delivery (Fig. 4a, two-sided likelihood ratio test, $P = 0.01$, see Supplementary Table 6) and, using this relationship, we back-calculated methane emissions to the pre-1940s natural baseline (Eq. (15)). Accordingly, prior to the intensification of agriculture, streams would have emitted methane over a range of 0.1 to 1.0 mmol $CH_4$ per $m^2$ per day (first and third quartile, respectively, Fig. 4b). Today, post intensification of agriculture, while the overall range in methane emissions is similar at 0.4 to 1.6 mmol $CH_4$ per $m^2$ per day (first and third quartile, respectively, Fig. 4c), just as for organic matter (Fig. 1c vs. d), the median has moved markedly to the right and, overall, average methane emissions have increased 3.5-fold (0.2 mmol $CH_4$ per $m^2$ per day vs. 0.7 mmol $CH_4$ per $m^2$ per day). At the category

level, methane emissions from streams under severe sediment pressure have increased sevenfold, while those under mild pressure threefold (on average, median to median, Eq. (16)).

## Discussion
Headwater streams are recognized sources of methane[12,40] yet it is unclear whether that methane is part of a natural healthy ecosystem or one degraded through the ingress of excess sediment from intensive agriculture[11,20]. Here, by defining a modern natural baseline we were able to systematically assess the effects of excess fine-sediment delivery across a wide range of streams and, thus, separate natural from human-enhanced methane emissions.

Methane is predominantly produced in the bed of headwater streams[4,19] and correlations between either methane production or concentration and sediment organic matter are recognized[20,24,38,41]. However, systematic partitioning of the relative influence of organic matter and temperature on methane production is lacking, a clear gap in our understanding as our climate continues to warm[31]. Here, by standardizing sediment methane production to 15 °C, we were able to isolate the effect of organic matter from that of temperature and demonstrate a very large variation (~10,000) in methane production. We can set our observations for streambed sediments into a wider context by comparing them to lake sediments and wetland soils that are the recognized largest freshwater sources of methane[42]. Published data (see Supplementary Fig. 3) shows that the potential control of methane production by organic matter is widespread in stream sediments[20,43,44], lake sediments[37,45–49], and wetland soils[49–52]. Moreover, the sensitivity of methane production to organic matter, i.e., the increase in methane production per tenfold increase in organic matter (%), is comparably high for both streambed sediment and wetland soils, with a common 38-fold increase in methane production per tenfold increase in organic matter, which is sufficient to drive our observed 10,000-fold variation in streambed methane production.

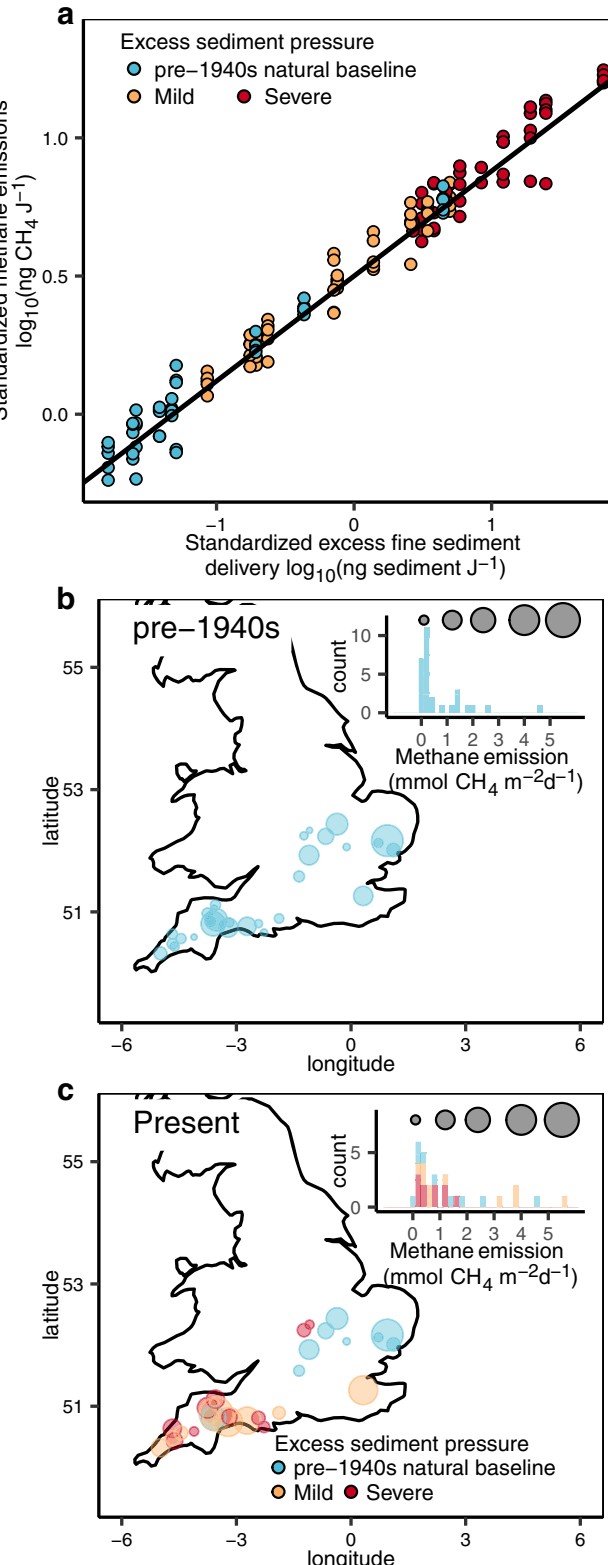

**Fig. 4 Excess fine-sediment delivery to streams increases methane emissions ($n = 142$ observations in 29 streams). a** Positive correlation (two-sided likelihood ratio test, $P = 0.01$) between standardized methane emissions from streams and standardized excess fine-sediment delivery to streams. The standardized excess fine-sediment delivery in the pre-1940s natural baseline streams, just as in Fig. 1b, represents the natural hydrological baseline. **b** Map of methane emissions back-calculated to the pre-1940s and **c**, map of present-day methane emissions from streams. Note the standardized methane emissions (y axis) data in **a**, are partial residuals while holding the excess sediment pressure at the median value to illustrate its overall relationship with the standardized delivery of excess fine sediment in each category (x axis, see Supplementary Table 6 for model selection and more details therein). Histograms in **b**, **c** give the distribution of methane emissions, and the size of circles the estimated rate of methane emissions.

Note that this 1.8-fold increase also assumes a simple—immediate—physiological response by methanogens to warming, whereas we have also demonstrated disproportionate increases in methane emissions after long-term (2006–2017[32]) warming of 4 °C in experimental ponds. For example, rather than the 1.8-fold higher emissions predicted for a physiological response to 4 °C of warming, we previously measured 2.4-fold and ascribed that disproportionate increase to stronger hydrogenotrophic methanogenesis in response to long-term warming[32]. Therefore, any further warming will initially increase streambed methane production predictably with temperature, just as we saw after 1 year of warming in our ponds and others witnessed after warming peat[53]. With more substantial long-term warming, however, there is every chance that disproportionate increases will emerge though they will still remain relatively modest compared to increases in organic matter.

Furthermore, the pronounced effect of organic matter on streambed methane production is even more evident if we look to the past. Here, we use the 1940s as a common baseline to compare the effects of both temperature and organic matter. Since the 1940s, average temperatures have increased by ~0.7 °C in England[54], corresponding to a 1.1-fold increase in methane production for the 1.1 eV reported here[32]. Nevertheless, over the same time frame, the increase in agriculturally derived streambed organic matter from 23 to 100 g per m$^2$ would have driven a 100-fold increase in methane production, dwarfing the effects of warming to date.

At a far larger catchment scale, circa 3000–30,000 km$^2$ vs. <15 km$^2$, here, fluvial methane emissions have been shown to correlate with catchment net ecosystem production (NEP)[41]. By definition, catchment NEP represents the organic carbon available for export that will contribute part of the organic matter (allochthonous carbon) driving natural methane production on the bed[4]—even if, at the smaller-scale, catchment NEP is comparatively constant[55]. We acknowledge that a correlation between methane emissions and organic matter in agriculturally impacted rivers has been reported previously[20,24,38], however, formal characterization of the relationship between organic matter, fine sediment, and stream methane has been lacking to date. Here, in streams under excess fine-sediment pressure the natural NEP background is being augmented by excess organic matter derived from agricultural activity, increasing methane production above natural background rates. Any excess fine sediment from agriculture will, however, only impact streambed function if deposited, which, in turn, is regulated by stream power[27]. Here we selected 236 UK streams covering a wide range in discharge, channel slope, and bankfull width that, combined, represent a 40,000-fold variation in stream power. By standardizing excess

Here, the temperature sensitivity of methane production was conserved at 1.1 eV across all streams, regardless of organic content, and comparable with the 0.93 eV derived using data from many other aquatic, wetland, and rice-paddy systems[30]. Assuming that current streambed organic matter content remains constant, using 1.1eV, we would predict a further 1.8-fold increase in streambed methane production in response to the 4 °C warming scenario for the end of the twenty-first century[31,32].

fine-sediment delivery and methane emissions to stream power, we were able to account for variation driven by geomorphology and hydrology, and thus to isolate natural from human-enhanced methane emissions.

There is large variation in the magnitude of methane emissions from streams and rivers globally—be they natural or driven by the impact of human activities[12,41]. A recent meta-analysis of methane emissions from more than 500 streams and rivers revealed an average (diffusive) rate of 0.8 mmol $CH_4$ per $m^2$ per day (median, with first and third quartiles of 0.5 and 5.1 mmol $CH_4$ per $m^2$ per day, respectively)[2], which agrees very well with our present-day estimate of 0.7 mmol $CH_4$ per $m^2$ per day (median, with first- and third quartiles of 0.4–1.6 mmol $CH_4$ per $m^2$ per day). While our entire analysis is restricted to diffusive methane emissions, we do not believe our conclusions about the effects of fine sediment would be affected by ebullition (the rapid release and export of methane in bubbles[56]). First, methane production and any subsequent diffusive and ebullitive methane emissions increase in proportion to each other[32,57]. Second, our own recent high-resolution and direct measurement of methane emissions from headwater streams showed that <1% of total methane emissions were ebullitive[4], though the fraction may be greater in larger rivers[56].

More importantly, after separating natural from human-enhanced methane emissions we would expect natural stream emissions to be appreciably lower at 0.2 mmol $CH_4$ per $m^2$ per day (median, with first- and third quartiles of 0.1 to 1.0 mmol $CH_4$ per $m^2$ per day). While the overall range in methane emissions today is comparable to the natural baseline, the widespread intensification of agriculture has reduced the frequency of natural streams, shifting the distribution in emissions to the right and increasing stream methane emissions overall. As natural stream methane emissions are equivalent to ~30% of their present-day emissions—while challenging—it is at least possible that current emissions could be reduced by some 70% through management strategies aimed at eliminating the export of excess fine sediment to streams from agricultural catchments. Given the forecasts for more frequent rainfall extremes and concomitant elevated soil erosion and sediment loss rates, these issues likely mean such strategies need to include major structural land cover change to reduce off-site impacts of agriculture moving forward[25,26]. Furthermore, given that intensive agriculture is a global issue[29], methane emissions from streams and rivers (2.6 Tg $CH_4$ per year[2]) around the world could be reduced if such effective management is widespread.

In the future, synergy between excess organic matter and climate warming will continue to enhance streambed methane production. While warming is gradual and hard to reverse, controlling the run-off of fine sediment from land to streams is at least within our more immediate control. We suggest that management aimed at mitigating further increases in methane production fueled by excesses of organic matter are urgently needed, especially since current on-farm sediment control has been shown to be of limited effect[58].

## Methods

**Study site selection**. In total, fine-sediment delivery and streambed organic matter were estimated in 236 agricultural catchments across England and Wales that were representative of a range of agricultural activities. First, 182 catchments were selected from the 12,447 sites within the Environment Agency River Habitat Survey (RHS) database. Total (i.e., organic and inorganic) fine-sediment load (kg ha$^{-1}$ year$^{-1}$) from the upstream catchments were modeled using PSYCHIC, a process-based model of fine-grained sediment delivery in surface run-off or drain flow from agricultural land[59]. Any sites with a substantial influence from urban areas or sewage effluent were eliminated via screening to isolate the effect of agricultural land use. All sites were upstream of any lakes and reservoirs and were on independent watercourses. Based on map-based physical variables, namely catchment geology, distance from source (km), altitude (m above sea level), and

stream slope (m km$^{-1}$), sites were further selected to cover the range of natural stream types. The boundary values for this stream typology were loosely based on the physical characteristics associated with the seven RIVPACS IV super end groups, summarizing the range of biological stream types found in the UK[60]. Full details regarding the site selection process are given in ref. [61]. In addition, a further 54 sites were selected according to the extent of participation in agrienvironment schemes in their catchment to give 236 in total (for details, see ref. [62] and see Supplementary Table 1 for the statistical summary of geomorphological and hydrological data for the 236 study streams). The positions of the 236 study streams were visualized in Fig. 1 on maps using the map_data() function in the R package ggplot2 (version 3.0.0)[63].

**Excess fine-sediment delivery into streams above unimpacted baselines**. Present-day sediment delivery to the 236 streams was computed using PSYCHIC but critically, with corrections for the impacts of the current uptake of on-farm best practice for water-quality protection driven by regulation, incentivization (e.g., agrienvironment schemes), and advice[28].

Lakes naturally trap sediments and their sediment accumulation rates are proportional to the sediment delivery to streams from the catchment[11]. Therefore, historical sediment delivery rates to streams were estimated from the sediment accumulation rates in lakes. In the UK, we would need to go back some 1000 years to establish truly "natural" rates of sediment yields from catchments to rivers[64]. Here, as the majority of dramatic modern increases in UK sediment yields occurred after the 1940s, we have used estimates of pre-1940s sediment yields to define a modern natural baseline representative of sediment inputs able to support a healthy aquatic ecology[11]. The 236 study streams were subsequently categorized into three groups based on the gap between the present rate of fine-sediment delivery and the historical natural baseline, i.e., the magnitude of present sediment delivery from agricultural sources in excess of their pre-1940s baselines—from here, the natural baseline. If the modeled rates of fine-sediment delivery were below their estimated natural baselines, streams were categorized as being natural and under good ecological status without any excess fine-sediment pressure. If the modeled delivery of fine sediment was greater than their estimated natural baselines, the pressure of excess fine sediment was categorized as mild and some mitigation is required. Finally, if the modeled delivery of excess fine sediments was greater than the upper thresholds of estimated natural baselines, i.e., the maximum sediment inputs a stream can tolerate and still sustain a healthy aquatic ecology, the pressure of excess fine sediment was categorized as severe and a targeted mitigation plan is urgently required.

**Streambed organic matter**. Organic matter deposited on the 236 streambeds was measured as per ref. [27] using the disturbance technique. An open-ended, stainless-steel cylinder (height 75 cm; diameter 48.5 cm) was carefully inserted into an undisturbed patch of streambed to a depth of at least 10 cm, until an adequate seal with the substrate was achieved, and the depth of water within the cylinder measured. The streambed within the cylinder was then disturbed to a depth of ~10 cm, vigorously agitated for one minute to suspend any surface and subsurface fines, and a pair of 50 ml samples quickly taken. For each stream reach sampled, samples from four locations (2 erosional, 2 depositional) were collected in order to characterize the reach-scale average[65]. Samples were collected during low to medium flows and no samples were collected during or immediately after peak flow events.

The samples were refrigerated and kept in the dark until being analyzed within one week of return to the laboratory. The samples were passed through a 2 mm sieve, to remove leaves and twigs, prior to filtration using pre-ashed, washed and dried 90 mm Whatman Glass Microfibre GF/C filters (pore size 1.2 µm). Organic matter content (expressed as ash-free-dry weight, AFDW, in g m$^{-2}$) in the filtered samples was derived from loss on ignition (LOI) by drying in a pre-heated oven at 105 °C overnight and ashed in a pre-heated muffle furnace at 500 °C for 30 min. We recognize that the clay content of sediments can influence the accuracy of LOI-determined organic content[66]. Here, however, as the clay content of our sediments was consistent across the 236 samples (first and third quartiles, 9% and 14% clay by dry weight, respectively), we ignored the 0.5% correction to our organic carbon content estimates due to the 5% variation in clay content[66]. Averaging the four samples provided an effective measure of deposited fine sediment at the reach scale which has been shown to be reliable across a wide range of stream types (>60% boulders and cobbles to >60% sand and silt) and not affected by operator bias[65].

**Standardization of excess delivery of fine sediment**. The mass of sediment deposited on a streambed is regulated by the balance between sediment inputs and a stream's transport capacity to flush out sediment[27]. Specific stream power ($\omega$, in W m$^{-2}$) is used as an index of the capacity of a stream to carry sediment and can be calculated using median annual maximum flow, channel slope, and bankfull width as per ref. [27]:

$$\omega = \rho g Q_{MED} S / W_{BF} \qquad (1)$$

where $\rho$ is the density of water (kg m$^{-3}$), $g$ is the gravitational acceleration (m s$^{-2}$), $Q_{MED}$ is the median annual maximum flow (m$^3$ s$^{-1}$), $S$ is the channel slope (m m$^{-1}$) and $W_{BF}$ is the bankfull width (m). Specific stream power, with the unit of J m$^{-2}$ s$^{-1}$,

represents the flux of kinetic energy per unit area of streambed per unit of time that is liberated from the potential of a stream water mass moving down a channel slope. This can be interpreted as the energy within a flowing stream to drive, for example, erosional processes and re-suspension of streambed sediments.

In order to quantify the balance between excess delivery of fine sediment into a stream and the stream's transportation capacity, we standardized any excess delivery of fine sediments to specific stream power as:

$$SD = (D + 1)/\omega \qquad (2)$$

Where $D$ is the excess delivery of sediments to streams above the pre-1940s natural baseline (ng m$^{-2}$ s$^{-1}$). Therefore, SD, the standardized delivery of excess fine sediment, in units of ng J$^{-1}$, represents the excess fine-sediment delivery standardized to each joule of stream flow energy. As the sediment delivery in natural streams, i.e., those that are not under excess sediment pressure, is defined as 0, $D + 1$ was used here to avoid any missing values after $\log_{10}$ transformation in the following correlation analysis (see statistical analysis below). Thus, the $\log_{10}$ transformed standardized excess sediment delivery—in natural baseline streams—represents a natural hydrological baseline that can be used to separate out the human-induced streambed organic matter (Eq. (8)) or methane emission (Eq. (14)) in the study streams.

**Sediment collection and laboratory incubations.** Sediments were collected from 14 other streams in southern England independent of the 236 excess sediment database (see Supplementary Fig. 1 for distribution of the study sites and Supplementary Table S1 for stream names, sampling time, stream geology types, and particle sizes). In 2013, sediments were collected from the main channels of three Chalk and three Greensand streams using small, hand-held corers (internal diameter 34 mm, polycarbonate) to quantify methane production. In the Wylye and the Avon, fine sediments that collect under the dominant macrophyte (*Ranunculus* sp.)[67] or in the channel margins were also collected to quantify the effect of different sediment patch types e.g., plant, marginal and main-channel patches (five cores per patch type per stream). As the sediments from plant patches had the strongest methane production capacity, the study was subsequently extended, in 2016, by collecting fine sediments from plant patches in another six Chalk and two Greensand streams using the same techniques (three or four cores per stream), to investigate methane production both with and without additional methanogenic substrates. The sediments were kept intact in their corers at 4 °C in the dark before handling in an anoxic glove box in the laboratory the next day.

Sub-samples (~3 g) of the bottom 3–5 cm of sediment cores were transferred into 12-ml gas-tight vials (Labco Exetainer®, Lampeter, UK) in an anoxic glove box (CV204, Belle Technology, Portesham, UK). Water from each sampling site was flushed with oxygen-free nitrogen (N$_2$, BOC, Guildford, UK) for 10 min and the deoxygenated water (4 ml) added to each vial. For the experiment with additional substrates, sodium acetate (Sigma-Aldrich®, for molecular biology), sodium propionate (Sigma-Aldrich®, for molecular biology), betaine (perchloric acid titration, ≥98% purity, Sigma-Aldrich®), hydrogen (research-grade, BOC, Guildford, UK) and trimethylamine (TMA, Sigma-Aldrich®) were used as their potential utilization by methanogens is well characterized[33]. Except in the case of hydrogen, deoxygenated water (3.6 ml), as well as the substrate stock solution (0.4 ml, 100 mM) were added to each vial to create final concentrations of 10 mM for each substrate and the vials were then sealed. For hydrogen, 1 ml of pure hydrogen was injected through the septum into each vial using a gas-tight syringe (1 ml, Hamilton) to create a concentration in the headspace of ~17% v/v. A further set of vials were left unamended as controls. All the prepared vials were then placed in temperature-controlled incubators covering a range from 5 to 26 °C in ~5 °C increments and incubated for up to 4 days.

The production of methane was quantified every 24 h by withdrawing 100 μl gas samples from the headspace of each vial and injecting these into a gas chromatograph fitted with a flame-ionizing detector (GC/FID, Agilent Technologies UK Ltd., South Queensferry, UK) as per ref. [32]. Concentrations of methane were calculated from peak areas calibrated against standards (prepared by diluting pure methane, BOC, Guildford, UK). The total amount of methane in each vial (headspace and dissolved in the water) was calculated using published solubility coefficients for methane[68].

**Characteristics of incubated sediments.** Sediment samples were oven-dried and particle size analysis was carried out by hand using sieves (Endecott Ltd, UK) of various sizes (16, 13.2, 8, 4, 1.4, 0.5, 0.25, 0.125, 0.063, and <0.063 mm). Each size fraction was weighed separately and the median particle size was determined. Organic carbon content of the incubated fine-sediment (<2 mm) samples was determined directly by elemental analysis (Sercon Integra2) after removing inorganic carbon using 1M HCl as per ref. [69] and then converted to combustible organic matter, i.e., ash-free-dry weight as per ref. [66] to align with the UK-wide excess fine-sediment survey.

**Methane emissions from streams.** To test for any relationship between methane emission and excess fine-sediment delivery, water samples ($n = 5$) were collected from the middle of the main channel of a subset of 29 streams from across our three categories of excess fine-sediment pressure ($n = 9$ for pre-1940s natural baseline, $n = 10$ for mild and $n = 10$ for severe of the total 236 streams) in August

2020 (the positions of the 29 streams were visualized on maps in Fig. 4 using the map_data() function in the R package ggplot2[63], see also Supplementary Fig. 1 for more details). The water sample once collected was discharged immediately into a gas-tight 12 ml gas-tight vial (Exetainer, Labco) and allowed to overflow three times before being fixed using 100 μl ZnCl$_2$ (50% w/v). Once back in the laboratory, methane concentrations in stream water were quantified by headspace equilibration as per ref. [19]. A headspace of 2 ml was created by introducing analytical grade helium using a two-way valve and gas-tight syringe. After equilibration for 24 h, gas samples of 100 μl was withdrawn from the headspace and injected into a gas chromatograph fitted with a flame-ionizing detector (Agilent Technologies). Methane emission (ME) was subsequently calculated using the following equation:

$$ME = k_{CH4} \times ([CH_4] - [CH_{4(sat)}]) \qquad (3)$$

Where ME is methane emission (ng CH$_4$ m$^{-2}$ d$^{-1}$), [CH$_4$] is the measured concentration of CH$_4$ in stream water (ng CH$_4$ m$^{-3}$) and [CH$_{4(sat)}$] the methane concentration at atmospheric equilibration (ng CH$_4$ m$^{-3}$) calculated using an atmospheric concentration of 1.8 ppm[31] and the solubility of methane at stream temperature. The stream water temperature was measured in 12 out of the 29 streams. As the average concentration of methane in stream water varied by 200-fold (from 5 nM to 1300 nM) while the in situ water temperature varied by only 1.3-fold (from 13.0 °C to 17.3 °C), the historic temperature data in August (download from Water Quality Archive provided by Environment Agency, https://environment.data.gov.uk/water-quality/view/landing) were used as conservative estimates of methane emission where in situ temperature data were missing. $k_{CH4}$ is the gas transfer velocity for methane (m d$^{-1}$) derived from $k_{600}$ (m d$^{-1}$) as follows[70]:

$$k_{600} = 4725 \times (VS)^{0.86} \times Q_{50}^{-0.14} \times D^{0.66} \qquad (4)$$

$$k_{CH4} = k_{600} \times (ScCH_4/600)^{-0.5} \qquad (5)$$

Where $V$ is the stream velocity (m s$^{-1}$), $S$ is the slope (m m$^{-1}$), $D$ is the water depth (m) and $Q_{50}$ is the median flow (m$^3$ s$^{-1}$) estimated from $Q_{MED}$ in Eq. (1) based on their relationship fitting a median regression using the discharge data from UK National River Flow Archive ($n = 536$, Supplementary Information Fig. 4 and discussion therein). ScCH4 is the Schmidt number for methane at either in situ or historical temperature.

To explore any correlation between methane emission and excess fine-sediment delivery, methane emission was standardized to the same unit of stream flow energy, as for excess fine-sediment delivery, in Eq. (2) using:

$$SME = ME/(\omega \times 86400) \qquad (6)$$

Where ME is methane emission (ng CH$_4$ m$^{-2}$ d$^{-1}$, see Eq. (3)) and $\omega$ is specific stream power (J m$^{-2}$ s$^{-1}$). The constant, 86400, is seconds per day. SME is standardized methane emissions in units of ng CH$_4$ J$^{-1}$ and thus represents methane emissions standardized to each joule of stream flow energy.

**Statistical analysis**

*Quantifying excess fine-sediment delivery and streambed organic matter.* To quantify the effect of excess fine-sediment delivery on streambed organic matter, a linear model was fitted to the streambed organic matter data for the 236 streams (see Fig. 1b) in the form:

$$\log_{10}OM_i(SD) = slope \times \log_{10}SD_i + intercept + \varepsilon_i \qquad (7)$$

Where $\log_{10}OM_i(SD)$ is the $\log_{10}$ scale of streambed organic matter in any stream $i$ ($i = 1, 2, \ldots, 236$). $\log_{10}SD_i$ is the $\log_{10}$ scale of standardized excess fine-sediment delivery in stream $i$. The magnitude of "excess fine-sediment pressure" (category e.g., pre-1940s natural baseline, mild, severe) was modeled as an interactive term. See Supplementary Table 3 for model selection procedures.

The organic matter in streams under mild or severe excess sediment pressure was lower in the 1940s before any excess fine-sediment delivery increased. The increase in streambed organic matter, due to excess fine-sediment delivery, and the streambed organic matter back in the 1940s were calculated using the standardized excess fine-sediment delivery in natural baseline streams:

$$\log_{10}OM_{i,1940s} = \log_{10}OM_{i,present} - 0.36 \times \Delta\log_{10}(SD) \qquad (8)$$

$$\Delta OM = OM_{mean,present}/OM_{mean,1940s} \qquad (9)$$

Where $\Delta\log_{10}(SD)$ is the difference in standardized excess fine-sediment delivery (on a $\log_{10}$ scale) in streams under mild or severe fine-sediment pressure compared with that in natural baseline streams, i.e., without any excess sediment pressure. $\log_{10}OM_{i,present}$ and $\log_{10}OM_{i,1940s}$ are the $\log_{10}$ scale of organic matter currently in any stream $i$ ($i = 1, 2, \ldots, 236$) or back to the 1940s. The constant, 0.36, is the slope of the relationship between streambed organic matter and standardized excess sediment delivery in Eq. (7). By reversing the $\log_{10}OM_{i,1940s}$, the real organic matter on streambeds, i.e., $OM_{1940s}$ was back-calculated and $\Delta OM$ is therefore the increase in average streambed organic matter in mildly or severely impacted streams, i.e., $OM_{mean,present}$, relative to the natural baseline, i.e., $OM_{mean,1940s}$.

*Capacity and temperature sensitivity of streambed methane production.* As we collected streambed sediments at different times of the year, our data were

unbalanced. To derive overall estimates for the capacity and temperature sensitivity of methane production across streams, linear mixed-effects models were used to account for the variance among sample collection date across streams[71]. According to the Boltzmann–Arrhenius equation, we estimated the capacity and temperature sensitivity of methane production according to:

$$\ln F_{ij}(T) = (\overline{E_{MP}} + a_{ij})\left(\frac{1}{kT_{15}} - \frac{1}{kT_{ij}}\right) + \left(\overline{\ln F(T_{15})} + b_{ij}\right) + \varepsilon_{ij} \quad (10)$$

Where $\ln F_{ij}(T)$ is the natural-log-transformed rate of methane production by any sediment sample collected from any stream $i$ ($i = 1, 2, …14$) in month $j$ ($j = 1, 2, …, 12$). $T$ is the specific incubation temperature (K) and $T_{15}$ is the mean temperature (288.15K, i.e., 15 °C) across all incubations. The term, $\frac{1}{kT_{15}}$, is used to standardize the plot and the term, $\overline{\ln F(T_{15})}$, represents the average methane production capacity at the mean temperature (15 °C) and $k$ is the Boltzmann constant ($8.62 \times 10^{-6}$ eV K$^{-1}$). The slope, $\overline{E_{MP}}$, is the estimated overall temperature sensitivity of methane production, here expressed as apparent activation energy in units of eV. Sample dates and individual streams were included as random effects on the slope ($a_{ij}$) and intercept ($b_{ij}$) to account for the variation among streams on each sampling date. $\varepsilon_i$ is the unexplained error with an assumed normal distribution $N$ (0, $\sigma^2$).

We fitted the Boltzmann–Arrhenius equation to the data using mixed-effect models using the "lmer" function in the "lme4" package (version 1.1-23)[71] of R statistical software (version 4.0.0)[72]. Model selection was performed using a top-down strategy (see the model selection procedures in Supplementary Table 4). Confidence intervals at the 95% level of probability (95% CI) were calculated using the "confint" function provided in the "lme4" package.

*Characterizing the relationship between sediment organic matter and methane production capacity.* To assess the effect of organic matter on methane production capacity, a linear model was fitted to the methane production capacities of each of the 14 streams in our study according to:

$$\ln MG_i(OM) = 7.2 \times \log_{10}(OM_i) - 23.3 + \varepsilon_i \quad (11)$$

Where $\ln MG_i(OM)$ is the natural-log-transformed methane production capacity from stream $i$ ($i = 1, 2, …14$) derived from Eq. (10). $\log_{10}(OM_i)$ is the $\log_{10}$ scale of streambed organic matter and $\varepsilon_i$ is the unexplained error with an assumed normal distribution $N$ (0, $\sigma^2$). The constant, 7.2, is the slope of the relationship between methane production capacity and streambed organic matter while the other constant, -23.3, is the intercept for methane production capacity (in natural log scale) when streambed organic matter is minimal.

Using the slope of 7.2, we estimated the increase in methane production capacity:

$$\triangle MG = e^{7.2 \times (\log_{10} OM_{present} - \log_{10} OM_{1940s})} \quad (12)$$

Here, the $\Delta MG$ is the fold increase in methane production capacity and the constant 7.2 the slope defined by the linear relationship between streambed organic matter and methane production capacity in Eq. (11). $OM_{present}$ and $OM_{1940s}$ are the median value of streambed organic matter calculated from Eq. (8) (see the histogram in Fig. 1c, d for their distributions) of 100 g m$^{-2}$ and 23 g m$^{-2}$, respectively.

*Capacity and temperature sensitivity of streambed methane production with additional substrates.* Overall estimates of the temperature sensitivity and capacity of methane production across streams were determined using mixed-effects models as in the previous section:

$$\ln F_i(T) = (\overline{E_{MP}} + a_i)\left(\frac{1}{kT_{15}} - \frac{1}{kT_i}\right) + \left(\overline{\ln F(T_{15})} + b_i\right) + \varepsilon_i \quad (13)$$

Individual streams were the sole random effect included here ($a_i$, on the slope or $b_i$, on the intercept) as the sediments used for incubations with additional substrates were collected only once. The effect of additional substrates (i.e., acetate, hydrogen, betaine, etc.) on both the temperature sensitivity (slope) and capacity of methane production (intercept at 15 °C) was incorporated into the models as a fixed effect and model selection followed the same procedure as above (Supplementary Table 5).

*Correlation between standardized excess fine-sediment delivery and standardized methane emissions.* To quantify the effect of standardized excess fine-sediment delivery on methane emissions from streams, standardized methane emission data for 29 streams were fitted into a linear mixed-effect model of the form:

$$\log_{10} SME_i(SD) = slope \times \log_{10} SD_i + (intercept + b_i) + \varepsilon_i \quad (14)$$

Where $\log_{10} SME_i(SD)$ is the $\log_{10}$ scale of standardized methane emission in any stream $i$ ($i = 1, 2, …, 29$) and $\log_{10} SD$, the $\log_{10}$ scale of standardized excess fine-sediment delivery. As five replicates were taken from each stream to estimate methane emissions, a random-intercept only model was used to account for the variation within each stream ($b_i$) (see Supplementary Table 6 for model selection).

Similar to the stream organic matter (see Eqs. (8) and (9)), methane emissions in streams under mild or severe excess sediment pressure were lower pre-1940s before any modern-day excess fine-sediment delivery started. And again, the

increase in methane emissions due to the excess fine-sediment delivery and the methane emission back to pre-1940s can be back-calculated using the standardized excess fine-sediment delivery in natural baseline streams:

$$\log_{10} SME_{i,1940s} = \log_{10} SME_{i,present} - 0.38 \times \triangle \log_{10}(SD) \quad (15)$$

Where $\Delta\log_{10}(SD)$ is the difference in standardized excess fine-sediment delivery (on a $\log_{10}$ scale) in streams under mild or severe fine-sediment pressure compared with that in natural baseline streams, i.e., without any excess sediment pressure and was derived from the 236 streams originally sampled (see Eq. (8)). $\text{Log}_{10} SME_{i,present}$ and $\log_{10} SME_{i,1940s}$ are the $\log_{10}$ scale of standardized methane emissions in any stream $i$ ($i = 1, 2, …, 29$) either today or back to pre-1940s. The constant, 0.38, is the slope of the relationship between standardized methane emission and standardized excess sediment delivery in Eq. (14).

By holding stream power constant, real methane emissions in the 1940s can be calculated by reversing Eq. (6). And as the distribution of methane emissions was skewed (see Fig. 4b, c in the main text), the medians were used here to present the increase in methane emissions in present times relative to the pre-1940s natural baseline:

$$\triangle ME = ME_{median,present} / ME_{median,1940s} \quad (16)$$

$\Delta ME$ is the increase in median of methane emissions from mildly or severely impacted streams, i.e., $ME_{median,present}$, relative to the natural baseline i.e., $ME_{median,1940s}$.

**Reporting summary.** Further information on research design is available in the Nature Research Reporting Summary linked to this article.

## Data availability

Data generated in this study are provided in the Source Data file. Source data are provided with this paper.

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

## Acknowledgements

We thank I.A. Sanders for technical assistance and F. Shelley, P. Fletcher, L. Ouyang, and A. Ferronatto for help with fieldwork and A. Arnold, C.P. Duerdoth, J.F. Murphy, A. Hawczak, J.L. Pretty, and P.S. Naden for their help with quantifying organic matter and collecting stream water to estimate $CH_4$ emissions and Y. Si and D.B. Kinkel who

commented on a draft manuscript. This work was supported by the UKRI-NERC (UK Research and Innovation, Natural Environment Research Council, NE/J012106/1, M.T., C.M.H., and J.I.J.), the UKRI-BBSRC (Biotechnology and Biological Sciences Research Council, BBS/E/C/000I0330, A.L.C. and Y.-S.Z.), the Department for Environment, Food and Rural Affairs (Defra contract WQ0128, J.I.J. and J.F.M.) and the Welsh Government (Contract Lot 3, No. 183/2007/08, J.I.J. amd J.F.M.). We thank the UK National River Flow Archive for discharge data, Environment Agency Water Quality Archive (Beta) for historic stream water temperature data and Met Office Hadley Centre for temperature data in England (www.metoffice.gov.uk/hadobs). We thank all landowners and tenants for granting access to sampling sites and Queen Mary University of London for supporting Y.Z. with her PhD.

## Author contributions

M.T. and C.M.H. conceived of the original study with later contributions from Y.Z. Y.Z. and L.O. collected the sediment samples and performed the incubation experiments. A.L.C. and Y.-S.Z. provided the sediment pressure estimates. J.I.J. and J.F.M. quantified the streambed organic matter. Y.Z. and L.R. analyzed the data. Y.Z. and M.T. wrote the paper and all the authors commented on and edited the manuscript.

## Competing interests

The authors declare no competing interests.
