## [Peer Review File · Nature Communications]

Separating natural from human enhanced methane emissions in headwater streamsEditorial Note: This manuscript has been previously reviewed at another journal that is not operating a transparent peer review scheme. This document only contains reviewer comments and rebuttal letters for versions considered at *Nature Communications*.

REVIEWER COMMENTS

Reviewer #2 (Remarks to the Author):

My major concern with the previous version was how it emphasized the sensitivity of methanogenesis to substrate availability (“fine sediment”) as compared to temperature. I argued that it is trivial that it varies by many orders of magnitude in response to the amount of organic matter, but a lot less drastically in response to changing temperatures. Although the authors spend a substantial fraction of their response letter on defending that presentation ;-), they also revised the manuscript substantially in line with my suggestions. I find that it makes a lot more sense now. The title more accurately reflects the novelty of the manuscript. The comparison of actual fluxes in the past (1940s) and present is valuable. I miss some discussion of trajectories into the future. How are agriculture, erosion, sediment deposits, and eutrophication expected to change in the area? Will management keep them constant, resulting in a greater relative importance of temperature?

The discussion of the significance of the results in a bigger (global) context draws too far-reaching conclusions, beyond the conditions that the study represents. From the comparison of present and pristine conditions in a set of headwater streams in a rather narrow geographic areas with relatively modest contrasts in environmental conditions, the authors draw conclusions on how much methane emissions can be decreased globally by better management of agriculture. This is far beyond what can be supported by the data. Basically, the authors extrapolate the increase in methane emission in a set of UK headwater streams from pristine conditions in the 1940s until today, to argue that it is “feasible” to reduce global methane emission by similar proportions. I disagree that 1) UK conditions can be extrapolated this way, 2) that watersheds of headwater streams are representative of the watersheds of global agriculture, at all stream orders, and 3) that this is feasible at the suggested level. It is argued and partly implicit (line 350) that global management of runoff to streams to reach “pristine conditions” is “feasible”, in contrast to the mitigation of warming. Those are two enormous challenges (unfortunately, there are reasons to doubt that any of them is feasible), and I don’t find the comparison here meaningful.

Reviewer #3 (Remarks to the Author):

The work highlights the impact modern intensive agricultural practices are having on methane emissions from streams. The data shows a much stronger influence from land management practices than from changes in temperature. While even a natural unimpacted stream would be expected to contribute to methane emission, this data suggests that changes to agricultural land management i.e. improvements in land management practices that reduce soil erosion risk, would help reduce methane emission from streams and river, thus contributing to a reduction in global greenhouse gas emission. This demonstrates additional value of better agricultural land management beyond simply carbon sequestration.

This is a clearly articulated paper. Generally, the methodology is very detailed, with perhaps two very minor points to clarify (see under detailed comment). Data handling and interpretation is explained, and appropriate methods are used to combine and compare different datasets. Figures and tables are clearly explained and nicely presented. The conclusions reflect the presented data.

This paper represents a robust piece of work that provides an important contribution in its field and has wider relevance. I would recommend that it be published

Detailed comments

Ln 402, "The streambed was then disturbed to a depth of approximately 10 cm,...". I'm assuming this was within the cylinder, but would be good to make this clear.

Ln 408 to 416, for loss on ignition (LOI): While I don't think calcium carbonate content will have been an issue because you didn't go above 500°C, fine fraction content (by fine fraction I mean silt and clay content) does influence LOI results. Did you consider differences in fine fraction?

Response to reviewers' comments for NCOMMS-22-10340-T

“Separating natural from human enhanced methane emissions in headwater streams”

Reviewer #2 (Remarks to the Author):

My major concern with the previous version was how it emphasized the sensitivity of methanogenesis to substrate availability (“fine sediment”) as compared to temperature. I argued that it is trivial that it varies by many orders of magnitude in response to the amount of organic matter, but a lot less drastically in response to changing temperatures. Although the authors spend a substantial fraction of their response letter on defending that presentation ;-), they also revised the manuscript substantially in line with my suggestions. I find that it makes a lot more sense now. The title more accurately reflects the novelty of the manuscript. The comparison of actual fluxes in the past (1940s) and present is valuable.

Many thanks for this, we are glad you appreciate our revisions.

I miss some discussion of trajectories into the future. How are agriculture, erosion, sediment deposits, and eutrophication expected to change in the area? Will management keep them constant, resulting in a greater relative importance of temperature?

*The short answer to these points is that we do not know how each will vary across 236 UK streams in the future. As we say on the very last line of the discussion “We suggest that management aimed at mitigating further increases in methane production fuelled by excesses of organic matter are urgently needed, especially since current on-farm sediment control has been **shown to be of limited effect**⁶⁹” and prior to that “In the future, synergy between excess organic matter and climate warming will continue to enhance streambed methane production” – so these specific methane drivers will continue and/or worsen, see below. It is also well documented that >80% of England’s rivers (complicated data for UK as a whole) are still failing in terms of nitrate and phosphate and given the recognised groundwater legacy for nitrate - that predicts future increases in nitrate - few of these elements are likely to remain constant. While we feel a full discussion of each in relation to methane would be overly speculative, we do now introduce and cite work by some of our authors specifically in relation to excess fine sediment and soil derived P worsening under some future climate scenarios i.e., more extreme rainfall events and also make the point that current management strategies will likely not be sufficient and that we need to see serious structural land cover change put in place. See lines 50-54 in the introduction that we return to on lines 348-358 in our toned-down discussion (in line with the comment below and editors summary). Given that our number-based global prediction for reducing methane emissions from streams through fine-sediment control was deemed too speculative (below) we feel that any more than that outlined above would also be too speculative. As R3 says “The conclusions reflect the presented data” and we should stick to that.*

The discussion of the significance of the results in a bigger (global) context draws too far-reaching conclusions, beyond the conditions that the study represents. From the comparison of present and pristine conditions in a set of headwater streams in a rather narrow geographic areas with relatively modest contrasts in environmental conditions,

the authors draw conclusions on how much methane emissions can be decreased globally by better management of agriculture. This is far beyond what can be supported by the data. Basically, the authors extrapolate the increase in methane emission in a set of UK headwater streams from pristine conditions in the 1940s until today, to argue that it is “feasible” to reduce global methane emission by similar proportions. I disagree that 1) UK conditions can be extrapolated this way, 2) that watersheds of headwater streams are representative of the watersheds of global agriculture, at all stream orders, and 3) that this is feasible at the suggested level. It is argued and partly implicit (line 350) that global management of runoff to streams to reach “pristine conditions” is “feasible”, in contrast to the mitigation of warming. Those are two enormous challenges (unfortunately, there are reasons to doubt that any of them is feasible), and I don’t find the comparison here meaningful.

We appreciate the reviewer’s reservation about our global extrapolation and have toned down the statement from a quantitative prediction to a more generic statement and have also revised our use of the word feasible that can be taken to mean easily doable as well something being simply possible – for our UK work “As natural methane emissions are equivalent to only ~30% of present-day emissions – while challenging – it is at least possible that current emissions could be reduced by ~70% through management strategies aimed at eliminating the export of excess fine sediment from agricultural catchments. Given the forecasts for more frequent rainfall extremes and concomitant elevated soil erosion and sediment loss rates, these issues likely mean such strategies need to include major structural land cover change to reduce off-site impacts of agriculture moving forward^{25,26}. And wider impact Furthermore, given that intensive agriculture is a global issue²⁹, methane emissions from streams and rivers (2.6 Tg CH₄ per year(ref.2)) around the world could be reduced if such effective management is widespread.” See lines 344-358 the full context.

Some of the methane emitted from streams and rivers around the world will be due to ingress of excess fine sediment from the land – as we have systematically shown here and others (Crawford and Stanley etc.,) have highlighted in the US – so there is nothing contentious in this closing remark - we are merely making the point that this it is a widespread, global issue.

Reviewer #3 (Remarks to the Author):

The work highlights the impact modern intensive agricultural practices are having on methane emissions from streams. The data shows a much stronger influence from land management practices than from changes in temperature. While even a natural unimpacted stream would be expected to contribute to methane emission, this data suggests that changes to agricultural land management i.e. improvements in land management practices that reduce soil erosion risk, would help reduce methane emission from streams and river, thus contributing to a reduction in global greenhouse gas emission. This demonstrates additional value of better agricultural land management beyond simply carbon sequestration.

This is a clearly articulated paper. Generally, the methodology is very detailed, with perhaps two very minor points to clarify (see under detailed comment). Data handling and interpretation is explained, and appropriate methods are used to combine and compare different datasets. Figures and tables are clearly explained and nicely presented. The conclusions reflect the presented data.

This paper represents a robust piece of work that provides an important contribution in its field and has wider relevance. I would recommend that it be published

Many thanks, we are very pleased that you appreciate the quality of our work and support it being published.

Detailed comments

Ln 402, “The streambed was then disturbed to a depth of approximately 10 cm,...”. I’m assuming this was within the cylinder, but would be good to make this clear.

We have added the phrase “The streambed within the cylinder was then disturbed...” to line 413 in the Methods to make this absolutely crystal clear.

Ln 408 to 416, for loss on ignition (LOI): While I don’t think calcium carbonate content will have been an issue because you didn’t go above 500°C, fine fraction content (by fine fraction I mean silt and clay content) does influence LOI results. Did you consider differences in fine fraction?

We must admit that we had not considered the effect of fine fraction content on LOI estimates of organic carbon in the 236-stream UK survey. We have now, and this is detailed on lines 424 to 431 “as the clay content of our sediments was consistent across the 236 samples (1st and 3rd quartiles, 9% and 14% clay by dry weight, respectively), we ignored the 0.5% correction to our organic carbon content estimates due to the 5% variation in clay content (De Vos et al. 2005)”. That is, the linear correction model in De Vos et al of Total organic content (TOC) = $-0.1046 \text{ Clay} + 0.5936 \text{ LOI}$ ($r^2=0.98$) predicts an offset of 0.5% for our data.

We have also clarified the conversion between organic carbon content of the sediments used in the incubations (Fig. 2) that was measured directly by elemental analysis and the UK wide survey of organic matter in the 236 streams determined as AFDW by LOI (see lines 499 to 502). Hence, the UK wide survey based on LOI is really a relative measure of organic matter in fine sediments between streams with similar clay contents (Fig. 1), whereas the relationship between organic carbon and temperature determined in the lab is based on absolute organic carbon.

De Vos, B., B. Vandecasteele, J. Deckers, and B. Muys. 2005. Capability of loss-on-ignition as a predictor of total organic carbon in non-calcareous forest soils. *Communications in Soil Science and Plant Analysis* **36**: 2899-2921.

REVIEWERS' COMMENTS

Reviewer #2 (Remarks to the Author):

The new version adequately addresses the concerns I raised on the previous version/s.

Regarding trajectories into the future, I agree that quantification (even within the geographic region of the study) is difficult. I think your more general discussion on this issue is fine. I also find it very good (and necessary) that the global extrapolation is removed.

The text around line 350, saying that "...natural methane emissions are equivalent to only ~30% of present-day emissions ... current emissions could be reduced by ~70%..." should be clarified. I think it means that the natural background emissions from the streams is just 30% of their current emissions. Readers may interpret it more broadly at a more general level, including all industrial and other emissions.

Although the revisions have changed the major message of the study substantially (from a comparison of temperature vs sediment load effects to a focus on the anthropogenic contribution), I find the current manuscript sound – it has a different but still important message.

Reviewer #2 (Remarks to the Author):

The new version adequately addresses the concerns I raised on the previous version/s.

Thanks

Regarding trajectories into the future, I agree that quantification (even within the geographic region of the study) is difficult. I think your more general discussion on this issue is fine. I also find it very good (and necessary) that the global extrapolation is removed.

Good

The text around line 350, saying that "...natural methane emissions are equivalent to only ~30% of present-day emissions ... current emissions could be reduced by ~70%..." should be clarified. I think it means that the natural background emissions from the streams is just 30% of their current emissions. Readers may interpret it more broadly at a more general level, including all industrial and other emissions.

We appreciate the reviewer's point here but were merely trying to cut down on our repetitive use of "stream" but have revised the phrasing to "As natural stream methane emissions are equivalent to approximately 30% of their present-day emissions – while challenging – it is at least possible that current emissions could be reduced by some 70% through management strategies aimed at eliminating the export of excess fine sediment to streams from agricultural catchments". See lines 293 to 297.

Although the revisions have changed the major message of the study substantially (from a comparison of temperature vs sediment load effects to a focus on the anthropogenic contribution), I find the current manuscript sound – it has a different but still important message.

Great, thanks